# Knowledge on the Complementary Feeding of Infants Older than Six Months among Mothers Following Vegetarian and Traditional Diets

**DOI:** 10.3390/nu13113973

**Published:** 2021-11-08

**Authors:** Malgorzata Kostecka, Joanna Kostecka-Jarecka

**Affiliations:** 1Department of Chemistry, Faculty of Food Science and Biotechnology, University of Life Sciences, Akademicka 15, 20-950 Lublin, Poland; 2Independent Public Healthcare Center in Łęczna, Krasnystawska 52, 21-010 Łęczna, Poland; kostecka-joanna@wp.pl

**Keywords:** complementary feeding, nutrition knowledge, infants, vegetarian diet, vegan diets, animal-based foods

## Abstract

Solid foods should be introduced not later than the age of six months, regardless of whether the family adheres to a traditional, vegetarian, or vegan diet. The aim of this study was to compare the knowledge on the complementary feeding of infants older than six months among mothers who adhere to traditional and vegetarian diets and to identify problems that require the assistance of a dietician. A total of 251 mothers of children aged 10–12 months participated in the study. Only 10% of vegetarian mothers declared that they had placed their children on a vegetarian diet, whereas 36 mothers adhered to a lacto-ovo-vegetarian diet during complementary feeding in the first year of life. Mothers adhering to a traditional diet were characterized by lower levels of knowledge (4.1 ± 2.3 points) than vegetarian mothers (5.3 ± 2.1 points). Mothers following a traditional diet were more likely to indicate a higher than recommended number of daily meals (OR = 1.76; Cl: 1.31–1.97, *p* < 0.001). Significant differences were noted in the respondents’ adherence to the BLW method, which was more readily implemented by vegan (*p* < 0.05) and lacto-vegetarian (*p* < 0.05) mothers and was least popular among mothers following a traditional diet (OR = 0.81; CI: 0.66–1.23, *p* < 0.04). Vegetarian mothers unnecessarily delayed the introduction of gluten and potentially allergizing foods to the children’s diets, sometimes even past the age of 12 months.

## 1. Introduction

The number of families adhering to a vegetarian diet, including families where infants and small children are raised vegetarian, has increased in highly developed countries in the last two decades [1]. In Europe, the prevalence of vegetarianism is estimated at 2% in France, 3.2% in Poland, 9% in Germany, 10% in Sweden, and 12% in the United Kingdom [2]. Children have a higher demand for energy and nutrients than adults, but there are no clear guidelines on whether vegetarian and vegan diets are appropriate for children [3,4]. The German Nutrition Society (DGE) does not recommend a vegan diet for infants, children, or adolescents [3]. In turn, the U.S. Academy of Nutrition and Dietetics has stated that well-planned vegetarian, lacto-vegetarian, and lacto-ovo-vegetarian diets are healthy and nutritionally adequate in all stages of life, including during pregnancy and lactation [5]. The Italian Society of Human Nutrition (SINU) has approved the vegan diet for pregnant and breastfeeding women, as well as for infants and children [6]. The French-speaking Pediatric Hepatology, Gastroenterology, and Nutrition Group recommends that children raised on vegan and vegetarian diets should receive vitamin and mineral supplements to prevent nutritional deficits, which may be observed at all life stages [7]. The discrepancies in nutritional recommendations can probably be attributed to the general scarcity of research on vegetarian diets and their impact on the health and growth of children.

According to the revised position statement of the Polish Society for Pediatric Gastroenterology, Hepatology, and Nutrition (2021), infants and small children raised on a vegetarian diet require supplementation and should be monitored by a specialist. A vegan diet is allowed, provided that the child remains under medical supervision, supplements are incorporated in the diet, and the parents are informed that an unbalanced diet and the absence of supplementation can pose significant risks for the child’s health or even life [8].

Communication with the parents plays a very important role in families where infants and children are raised on vegetarian or vegan diets. Children should be provided with the essential nutrients during breastfeeding and complementary feeding [9]. According to current guidelines, solid foods should be introduced not later than the age of six months, regardless of whether the family adheres to a traditional, vegetarian, or vegan diet. Different foods should be introduced gradually to promote healthy eating habits [10].

The aim of this study was to compare the knowledge on the complementary feeding of infants older than six months among mothers who adhere to traditional and vegetarian diets and to identify problems that require the assistance of a dietician. The research hypothesis states that mothers following a vegetarian diet have less knowledge about the order in which complementary foods should be introduced to a child’s diet past the age of six months and that they will delay the introduction of animal-based foods.

## 2. Materials and Methods

### 2.1. Study Design and Participants

The parents of children aged 10–12 months who had introduced complementary foods in the children’s diet were invited to participate in the study. The inclusion criteria were the absence of metabolic disorders or other disorders that require an elimination diet or foods for special medical purposes and adherence to a vegetarian diet or a traditional diet in the family. The exclusion criteria were prematurity and low birth weight requiring nutritional support (enteral or parenteral nutrition) in the first weeks of a child’s life. The mothers participating in the study were patients of several pediatric clinics in Lublin city and Lublin county, and they were members of community groups for families following a vegetarian diet. During a control interview with the pediatrician, mothers were asked for consent to participate in the study, and anonymous participants were recruited for the study by posting invitations in social media groups dedicated to vegetarian diets.

All mothers completed the questionnaire voluntarily and independently without the participation of an interviewer. The completed questionnaires were verified for completeness to ensure appropriate quality standards. A total of 341 questionnaires were returned, and of those, 251 fully and correctly completed questionnaires qualified for the study.

### 2.2. Data Collection

The research tool was an original questionnaire developed by the authors based on the Infant and Young Child Feeding (IYCF) assessment [11] and Complementary Food Frequency Questionnaire [12,13], as well as a questionnaire that had been applied by the authors in a previous study [14]. Mothers who visited pediatric clinics could complete the questionnaire in paper form or online, whereas social media group users completed the questionnaire online.

In the first part of the study, mothers filled in a questionnaire containing 30 questions on breastfeeding (duration, frequency), introduction of solid foods, factors that influenced their decision to begin complementary feeding (the child’s readiness to expand the diet, motor coordination, upright posture, interest in new foods), types of solid foods and their frequency, cooking techniques, and the types and frequency of liquid consumption (especially liquids such as water, herbal tea, fruit tea, sweetened hot drinks, juice). In the second part of the study, the mothers were asked 20 short questions assessing their nutrition knowledge and familiarity with Polish feeding guidelines for infants, including the following:the recommended duration of breastfeeding,the number of milk-based meals recommended at 7–8 months and 12 months of age,portion size expressed in milliliters,the first complementary foods to be introduced in a child’s diet,changes in meal consistency adapted to the child’s age and abilities (semi-liquid, mushy, gritty, solid),introduction of animal-based foods and foods with allergizing potential (cow’s milk, gluten, honey, nuts),baby-led weaning method,the parent’s and the child’s role in the feeding process—the parent decides what, when, and where to feed the child, but the child decides whether he/she wants to eat and how much.

Twenty points could be obtained for correctly answering all knowledge questions. The participants’ knowledge was evaluated as low if the total score was below 8 points, moderate if the total score was 8–11 points, and high if the total score exceeded 11 points.

The study was conducted between January and August 2021 in the Region of Lublin in southeastern Poland. The respondents returned 251 correctly filled questionnaires. All mothers gave their informed consent to participate in the study, and they were informed about the purpose of the research. The questionnaires were completed anonymously by the participants without the researchers’ assistance or intervention.

### 2.3. Data Analysis

Categorical variables were presented as sample percentages (%), and continuous variables were expressed by median values and the interquartile range (IQR). The differences between groups were analyzed in the chi-squared test (categorical variables) or the Mann–Whitney test (continuous variables). Before statistical analysis, data were checked for normal distribution in the Kolmogorov–Smirnov test.

The odds ratios (ORs) and 95% confidence intervals (95% Cls) were calculated. The reference categories (OR = 1.00) included adherence to nutritional guidelines and the introduction of solid foods between 4 and 6 months of age. The ORs were adjusted for maternal age (years), maternal nutrition knowledge score (points), type of diet, and adherence to the baby-led weaning (BLW) method. The significance of ORs was assessed by Wald’s statistics. The results of all tests were regarded as statistically significant at *p* < 0.05. Data were processed in the Statistica program (version 13.1 PL; StatSoft Inc., Tulsa, OK, USA; StatSoft, Krakow, Poland).

## 3. Results

All of the respondents were women. The mothers differed in age, place of residence, education level, and the applied diet (Table 1).

Only 10% of vegetarian mothers declared that they had placed their children on a vegetarian diet without animal-based foods, whereas 36 mothers adhered to a lacto-ovo-vegetarian diet during complementary feeding in the first year of life. The remaining respondents were of the opinion that a traditional diet was most appropriate for children in the first 12 months of life.

In the first six months of life, an infant’s nutritional requirements are fully met by breastmilk or infant formulas. The surveyed mothers were assessed for their knowledge about the recommended duration of breastfeeding, the minimum recommended duration of breastfeeding, and the maximum recommended duration of breastfeeding. A significantly higher number of vegetarian mothers (regardless of diet type and age) were aware that the recommended duration of breastfeeding was six months (*p* = 0.001) and that the minimum recommended duration of breastfeeding was three months (*p* = 0.002). In turn, a significantly higher number of mothers adhering to a traditional diet (*p* < 0.05) were familiar with the maximum recommended duration of breastfeeding that should be adapted to the mother’s and the child’s needs after complementary feeding begins.

### 3.1. Knowledge about Polish Recommendations on the Complementary Feeding of Infants

Mothers adhering to a traditional diet were characterized by lower levels of knowledge (4.1 ± 2.3 points) than vegetarian mothers (5.3 ± 2.1 points). The group of respondents with the highest levels of knowledge comprised a significantly higher number of vegetarian mothers (13.4 ± 1.2 points) than mothers adhering to a traditional diet (12.3 ± 1.0 points) (Table 2). The knowledge about complementary feeding was highest among vegan (*p* < 0.5) and lacto-ovo-vegetarian mothers, and it was lowest among mothers adhering to an ovo-vegetarian diet (*p* < 0.05).

Significant differences were noted in the respondents’ knowledge about the optimal age for introducing solid foods, number of daily meals, and the parent’s and child’s role in the feeding process. According to the latest recommendations, parents should respond appropriately to signs of hunger and fullness in order not to overfeed or force children to eat. Infants aged 7–8 months should eat 5 main meals and 2 snacks, whereas 4–5 main meals and 1–2 snacks are recommended at the end of 12 months [8]. Mothers following a traditional diet were more likely to indicate a higher than recommended number of daily meals (OR = 1.76; Cl: 1.31–1.97, *p* < 0.001). On average, children aged 7–8 months were fed 6.7 meals per day by mothers adhering to a traditional diet and 5.4 meals per day by vegetarian mothers. Average portion size was similar in both groups at 185 ± 16 mL. The average number of daily meals served to children aged 9–12 months was lower (5.8) only in the group of mothers following a traditional diet. Animal-based foods, including cow’s milk, were introduced sooner by mothers adhering to a traditional diet (OR = 2.11; Cl: 1.74–2.36, *p* < 0.001). Mothers following a traditional diet introduced milk-based foods (cow’s milk, cottage cheese, cream cheese) at the average age of 10 months, and vegetarian mothers—at the average age of 11.5 months (baby formula, yogurt).

Significant differences were noted in the respondents’ adherence to the BLW method, which was more readily implemented by vegan (*p* < 0.05) and lacto-vegetarian (*p* < 0.05) mothers and was least popular among mothers following a traditional diet (OR = 0.81; Cl: 0.66–1.23, *p* < 0.04).

The majority of the respondents were of the opinion that the decision on the type and frequency of meals should be made by the parent, whereas vegetarian mothers were more likely to indicate that the child can decide whether and how much he/she wants to eat (*p* < 0.05).

The surveyed mothers significantly differed in their opinion on whether repeated administration of the same foods/flavors can influence a child’s attitudes to those food products. Significantly more mothers adhering to a traditional diet than a vegetarian diet were convinced that such practices do not affect a child’s eating habits (*p* < 0.05) and that repeated exposure to the same foods is not necessary (OR = 0.76; Cl: 0.62–0.91, *p* < 0.004). According to most vegetarian mothers and around 50% of mothers adhering to a traditional diet, each new food should be introduced individually at time intervals. The evaluated groups differed considerably in their opinions on the introduction of gluten and potentially allergizing foods. The vast majority of vegetarian mothers argued that foods containing cow’s milk proteins (CMPs) should be completely eliminated from the diet of children allergic to CMPs and that a provocation test should not be attempted in the first year of life. Gluten was introduced later than recommended by vegetarian mothers (at 34 ± 9 weeks of age on average) in comparison to mothers following a traditional diet (23 ± 6 weeks on average) (*p* < 0.05).

No significant differences were found in the respondents’ knowledge about feeding techniques and accessories at the beginning of complementary feeding (*p* > 0.05) or the liquids recommended between 6 and 12 months of age. The vast majority of mothers from all groups were familiar with the latest recommendations of the American Academy of Pediatrics and the Polish Society for Pediatric Gastroenterology, Hepatology, and Nutrition, which discourage the administration of fruit juice to children younger than 1 year.

### 3.2. Factors Associated with the Age Range for Introducing Complementary Foods

Solid foods should be incorporated into an infant’s diet based on nutritional recommendations, including the guidelines formulated by the Nutrition Division of the Polish Society for Pediatric Gastroenterology, Hepatology, and Nutrition.

Maternal diet significantly influenced the age at which complementary foods and meals were introduced to a child’s diet. Products other than vegetables and fruit were introduced significantly later by vegetarian mothers (the first solid foods to be introduced were: pumpkin, apples, German turnips, potatoes, and carrots). More than 50% of the respondents introduced eggs at the average age of 8–9 months, and 46 mothers did not introduce meat or meat-based products in the first 12 months of life. In most cases, fish (one serving per week) were incorporated into a child’s diet past the age of 7 months. Mothers following a traditional diet introduced animal-based foods significantly earlier than vegetarian mothers (*p* < 0.05): eggs at 20 weeks, meat at 23 weeks, and fish at 28 weeks of age on average, at an average amount of 1.4 servings per week.

Maternal knowledge and familiarity with complementary feeding recommendations influenced infant feeding practices (Table 3). Mothers with low levels of knowledge introduced solid foods significantly earlier (*p* < 0.05) than mothers with high levels of knowledge. In the group of respondents with higher levels of knowledge, vegetarian mothers, in particular vegan mothers (*p* < 0.05), delayed the introduction of solid foods (OR = 2.08; Cl: 1.62–2.46, *p* < 0.001) and significantly delayed the introduction of animal-based foods.

Adherence to the BLW method also affected the age at which complementary foods were introduced to a child’s diet. Mothers who followed the BLW method were more likely to delay the introduction of solid foods past the age of 6 months, and animal-based foods—past the age of 8 months. Mothers who did not adhere to the BLW method were less likely to delay complementary feeding (OR = 0.84; Cl: 0.72–1.03), although a tendency to introduce solid foods prematurely was not observed in this group.

## 4. Discussion

Vegan/vegetarian diets are gaining popularity in Western societies, and a growing number of children are raised by vegan mothers [15]. Children require more energy and nutrients per body weight unit than adults to ensure the normal growth and development of neural, endocrine, and immune systems.

In some vegetarian diets, sufficient intake of protein, fat, omega-3 fats, vitamin B12, vitamin D, iron, calcium, and zinc may be challenging [16,17]. In children older than 6 months who are raised on a vegetarian or a vegan diet, the hematologic symptoms of vitamin B12 deficiency can by masked by the high consumption of folates [15]. The content of vitamin B12 in breastmilk is directly correlated with serum B12 levels in breastfed infants [15,18], which is why breastfed children should be introduced to foods that are a rich source of vitamin B12 [19]. These products include fish, eggs, and dairy, which may be in short supply in the diets of children raised by vegetarian parents. The intake of foods rich in vitamin B12 was insufficient in the studied population, and the respondents had low levels of knowledge about vitamin B12 supplementation in children and the health risks associated with its deficiency. Only 17 mothers had administered methylcobalamin supplements upon the pediatrician’s advice. A systematic review demonstrated that vitamin B12 deficiency can occur in up to 45% of vegan infants [20].

Vegetarian and vegan diets for children contain nearly identical or even higher levels of iron than the traditional diet. However, plant-derived iron is less available than heme iron from animal-based foods. Heme iron has an estimated bioavailability of 20–30%, whereas the bioavailability of non-heme iron is only 2–5% [21]. In the surveyed population, most mothers following a traditional diet (64%) were of the opinion that meat and meat-based products are a source of iron in infant diets and that delayed introduction of meat can lead to iron deficiency. The vast majority of vegetarian mothers were aware that a vegetarian diet can be modified to increase the bioavailability of non-heme iron. The absorption of non-heme iron is facilitated by ascorbic acid (found in citrus fruit, strawberries, and kiwi fruit) as well as other organic acids (citric acid, malic acid, lactic acid, tartaric acid, carotenoids, retinol) that are found in fruit and vegetables [7,17].

According to the American Academy of Pediatrics and the World Health Organization, solids foods and complementary feeding should be introduced at the age of around 6 months [22]. Baby-led weaning (BLW) is a method that allows babies to control solid food intake by “self-feeding” from the start of their experience with solid food. Puree foods are skipped and solid foods that are appropriate in texture and size are offered [23]. The respondents were familiar with the BLW method, but mothers following a traditional diet were less likely to apply this method and resorted to the BLW approach only after the child had been introduced to complementary foods (at the average age of 8 months) or when the child was a picky eater. Vegan mothers introduced the BLW method at the average age of 5 months, where the first complementary foods were cooked pumpkin, potatoes, and broccoli. The first recommended foods in the BLW approach, both in traditional and vegetarian diets, include bananas, avocados, soft sweet potatoes, baked pumpkin, broccoli, and cooked lentils [24,25]. In the BLW method, the risk of choking, iron or zinc deficiency, and growth retardation is low; therefore, nutritional deficiencies are unlikely to occur [25,26,27]. However, more than 50% of the mothers adhering to a traditional diet were of the opinion that the BLW method could contribute to selective eating or nutritional deficiency.

Wholesome protein that contains all essential amino acids plays a very important role in the first year of life. In developed countries where various sources of vegetable protein are available, vegan diets can fully cover the protein demand of children and adolescents [28]. However, vegan infants who are not breastfed are at risk of protein deficiency [29]. In the studied population, more than 1/3 of vegetarian mothers and nearly 90% of vegan mothers argued that plant-based beverages such as rice milk or almond milk can be an alternative source of protein during complementary feeding. The health benefits of these beverages have not been fully elucidated, but the complete elimination of breastmilk or baby formulas containing cow’s milk in favor of plant-based beverages can lead to nutritional deficiencies, including protein deficiency and malnutrition [30,31,32]. Lacto-ovo-vegetarian or lacto-vegetarian diets that incorporate diverse plant-based foods meet the nutritional requirements of children older than six months and can be safely administered [33].

Nutrition and diet should be a part of pediatric care and prevention. Pediatricians caring for children with vegetarian or restrictive diets should monitor their physical development and dietary intakes, if necessary, in cooperation with an appropriately trained dietician [34].

### Strengths and Limitations

The choice of the surveyed population was a strength of this study. The studied women were mothers of children aged 10–12 months, which is a period during which new foods are incorporated into a child’s diet. The reliability of the results could be compromised if the study were conducted on parents of older (2- to 3-years-old) children. The information about the order in which solid foods were introduced or the age at which complementary feeding began could be inaccurate and incomplete after several months or years. The fact that the vegetarian mothers adhered to various plant-based diets, ranging from strictly vegan to lacto-ovo-vegetarian diets that are easier to balance, was also a strength of this study. The choice of the surveyed population supported the identification of differences in maternal knowledge and infant feeding practices among mothers following various diets. This is one of the first studies to compare parental knowledge on infant nutrition, and it sets the directions for future research in this field.

The main limitation of the present study, as well as our previous research, was that it involved only women since the questionnaires were not filled by fathers. As a result, paternal knowledge about complementary feeding and its influence on the children’s eating habits could not be evaluated.

## 5. Conclusions

Vegetarian mothers had higher levels of knowledge about complementary feeding recommendations for children aged 6–12 months.

Mothers adhering to a traditional diet introduced animal-based foods earlier and administered a higher than recommended number of daily meals to children aged 7–8 months.

Vegetarian and vegan mothers were more likely to introduce new vegetables and fruits with the use of the BLW method than mothers following a traditional diet, who were of the opinion that the BLW approach increases the risk of nutrient deficiency.

Plant-based diets that adequately meet children’s demand for protein could pose a challenge for vegetarian mothers. Not all vegetarian mothers are aware of the health risks associated with the introduction of plant-based beverages as the main source of protein in the child’s diet.

Vegetarian mothers unnecessarily delayed the introduction of gluten and potentially allergizing foods to the children’s diets, sometimes even past the age of 12 months.

Vegetarian diets that are diverse and have been approved by a dietician are safe for young children.

Mothers adhering to a traditional diet should introduce complementary foods to the diets of their children at 6 to 12 months of age according to Polish infant nutrition guidelines.

## Figures and Tables

**Table 1 nutrients-13-03973-t001:** Maternal factors.

Maternal Factors		N	%	*p*-Value
Age of mothers (average, years), mean (95% Cl)	18–25	30	11.9	0.04
	26–30	89	35.5	
	31–35	95	37.8	
	>35	37	14.8	
Place of residence, n (%)	Rural area	54	21.5	0.002
	Urban area	197	78.5	
Education level, n (%)	Primary school	7	2.8	0.001
	Secondary school	56	22.3	
	University	188	74.9	
Type of diet	Traditional	141	56.2	0.002
	Vegetarian, including	110	43.8	
	Lacto-ovo-vegetarian, 42 (16.7%)			
	Ovo-vegetarian, 20 (7.9%)			
	Lacto-vegetarian, 15 (6.0%)			
	Vegan, 33 (13.2%)			

**Table 2 nutrients-13-03973-t002:** Knowledge about Polish recommendations on the complementary feeding of infants among mothers adhering to different diets.

	Mothers with a Traditional Diet (n, %)	Vegetarian Mothers (n, %)	*p*-Value
Level of knowledge			
Low	23 (16.3)	5 (4.5)	0.001
Medium	42 (29.8)	29 (26.4)	ns
High	76 (53.9)	76 (69.1)	0.003

Level of knowledge: below 7 points—low; 8–11 points—medium; above 11 points—high; ns—not significant.

**Table 3 nutrients-13-03973-t003:** Odds ratios (95% confidence interval) of the relationships between maternal nutrition knowledge scores, maternal diet, and the introduction of solid foods before the age of 4 months and past the age of 6 months.

	Mean Age of Solid Food Introduction (Weeks)	Introduction of Solid Foods by the Age of 4 Months (Ref.: Introduction of Solid Foods between 4 and 6 Months of Age)	Introduction of Solid Foods by the Age of 6 Months, (Ref.: Introduction of Solid Foods between 4 and 6 Months of Age)
Maternal nutrition knowledge score, (n)			
Low (0–7 points)	14.7	1.60 ** (1.19; 1.88)	0.69 ** (0.34; 0.77)
Medium (8–11 points)	16.5	1.14 (0.82; 1.29)	1.11 (0.78; 1.27)
High (12–15 points)	19.3	0.81 * (0.7; 1.09)	1.84 *** (1.32; 2.16)
Type of diet			
Traditional	16.2	1.51 ** (1.12; 1.88)	1.21 (0.86; 1.59)
Vegetarian	18.1	0.61 ** (0.33; 0.89)	1.77 *** (1.26; 2.11)
Age of mothers (average, years),			
18–25	14.5	1.31 * (1.04–1.46)	1.03 (0.89–1.07)
26–30	20.2	0.98 (0.84–1.05)	1.06 (0.96–1.11)
31–35	17.6	1.09 (0.98–1.12)	1.04 (0.87–1.09)
>35	19.1	0.77 * (0.61–0.89)	1.29 * (1.11–1.39)
Adherence to BLW method			
Yes	18.5	0.74 * (0.68; 0.94)	1.27 * (1.03; 1.36)
No	17.3	1.17 (0.98; 1.27)	0.84 * (0.72; 1.03)

Data are statistically significant at: * *p* < 0.05; ** *p* < 0.01; *** *p* < 0.001.

## Data Availability

Due to ethical restrictions and participant confidentiality, data cannot be made publicly available.

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
