# Peer review of "Knowledge on the Complementary Feeding of Infants Older than Six Months among Mothers Following Vegetarian and Traditional Diets"

_nutrients, 2021, doi:10.3390/nu13113973_

Round 1

Reviewer 1 Report

Malgorzata and Joanna,

Your paper title is confusing and needs revising, especially as it is the aim of the study and also the research hypothesis e.g. Whose knowledge?

What about infants introduced to complementary feeding at less than 6 months? Also, one point of data collection.

There appears to be no reference to WHO infant feeding guidelines in this paper.ESPGHAN guidelines?

 A total of 251 fully  and correctly completed questionnaires were qualified for the study. How was this measured? Line 78-79.Participants were self selected which could have skewed data. Also as mothers completed questionnaire of children 10-12 months of age relying on mother's recall of feeding. Not requiring ethical approval for participation would not meet all countries standards.

Line 120 Be consistent with reporting data e.g Only 10% of vegetarian mothers declared that they had placed their children on a vegetarian diet whereas 36 mothers adhered.....

Table 1 needs revising including explanation of Traditional education level.

Line 128 "and the maximum duration of breastfeeding". Identify this maximum with reference. Usually do not state a maximum as it is maternal choice as to when breastfeeding is ceased (or infant). Minimal mention of infant formula use.Lines 150-156 need review, further explanation.

Line 176 'food' rather than product, be consistent.

Review discussion- Line 230 Reference 18 (30 years old), Line 236  How many visited paediatrician? Line 239 spelling and use of 'heme' ?  LIne 250 grammar,

Strengths and Limitations - this needs total review.Line 283"which is the appropriate age for the introduction of complementary foods" Reference.  Also review complete Conclusion, paying attention to grammar.

Funding  "the authors did research with the help of volunteers". Who were the volunteers?

Author Response

Dear Reviewer,

Thank you for presenting us with an opportunity to submit the revised version of the manuscript entitled ‘Knowledge on the complementary feeding of infants older than six months among mothers following vegetarian and traditional diets.

We greatly appreciate the time and effort taken by the Reviewer to review our manuscript. We have addressed all issues indicated in the review report, and we believe that the revised version will meet the Journal’s requirements. The manuscript was checked by a professional translator.

Please find attached our responses to the Reviewer’s comments. The manuscript has been corrected, and all changes are highlighted in blue font.

Yours sincerely,

Malgorzata Kostecka

Thank you very much for your insightful review. We greatly appreciate the time and effort taken to review our manuscript and we agree that the proposed changes will contribute to the improvement of our manuscript. We hope you will find our improvements appropriate and comprehensive.

Your paper title is confusing and needs revising, especially as it is the aim of the study and also the research hypothesis e.g. Whose knowledge?

The title has been modified and shortened; in the aim of the study and research hypothesis, the word “parents” has been replaced with “mothers” for greater consistency.

What about infants introduced to complementary feeding at less than 6 months? Also, one point of data collection.

The 6-month point was chosen because, according to the recommendations of ESPGHAN, AAP, WHO and the Polish Society for Pediatric Gastroenterology, Hepatology and Nutrition, this is the time after which each child should have supplementary meals introduced to the diet. Exclusive breastfeeding is recommended until the end of the 6th month, then, according to the recommendations, it is necessary to expand the diet for the sake of children’s health.

There appears to be no reference to WHO infant feeding guidelines in this paper.ESPGHAN guidelines?

Thank you for pointing this out. As suggested by the Reviewer, we have added a reference to the WHO recommendations (item 23). We have also referred to the recommendations of the American Academy of Pediatrics and the Polish Society for Pediatric Gastroenterology, Hepatology and Nutrition.

A total of 251 fully  and correctly completed questionnaires were qualified for the study. How was this measured? Line 78-79.Participants were self selected which could have skewed data. Also as mothers completed questionnaire of children 10-12 months of age relying on mother's recall of feeding. Not requiring ethical approval for participation would not meet all countries standards.

251 fully and correctly completed questionnaires were qualified for the study - 341 parents were invited to the study, all of them consented to participate in the experiment and complete an anonymous questionnaire. The questionnaires were returned and checked for completeness. Only the questionnaires containing answers to all questions were qualified for further research.

The mothers who completed the questionnaire relied on their memory of expanding the infant's diet and on their knowledge that they applied in practice. This was the purpose of the study. A similar experimental design had been used in a previous large-scale study on extending the diet of infants and feeding toddlers.

In our previous studies, we did not apply for consent to the bioethics committee because the obtained data do not allow us to identify the participants of the study in any way, they are entirely population-based and are not based on the history of the disease or obtaining biological material. All mothers were informed about the purpose of the study and expressed their consent to participate by completing the questionnaire.

77-80

82-85

Line 120 Be consistent with reporting data e.g Only 10% of vegetarian mothers declared that they had placed their children on a vegetarian diet whereas 36 mothers adhered....

Thank you for your remark. We would like to explain that only 10% of mothers declaring a vegetarian diet of any type declared that they had placed their children on a vegetarian diet without animal-based foods, whereas 36 mothers placed their children on a lacto-ovo-vegetarian diet.

141

Table 1 needs revising including explanation of Traditional education level.

Thank you for this suggestion. The Table has been revised and  has been changed.

Line 128 "and the maximum duration of breastfeeding". Identify this maximum with reference. Usually do not state a maximum as it is maternal choice as to when breastfeeding is ceased (or infant). Minimal mention of infant formula use.Lines 150-156 need review, further explanation.

What we meant here was mothers’ knowledge about the maximum recommended duration of breastfeeding. The word “recommended” has been added for greater clarity.

As suggested by the Reviewer, further explanation has been provided, based on Polish recommendations.

150

172-175

Line 176 'food' rather than product, be consistent.

The relevant correction has been made, here and throughout the manuscript.

200,202

Review discussion- Line 230 Reference 18 (30 years old), Line 236  How many visited paediatrician?

Line 239 spelling and use of 'heme' ? 

LIne 250 grammar,

As suggested by the Reviewer, more recent references have been cited (items 18 and 19).

Only 17 respondents declared that vitamin B12 supplementation was suggested during visits to a pediatrician.

Iron from food comes in two forms: heme and non-heme.

A different definition of the BLW method has been given.

254

275-276

Strengths and Limitations - this needs total review.Line 283"which is the appropriate age for the introduction of complementary foods" Reference.  Also review complete Conclusion, paying attention to grammar.

We agree that this passage was not clearly written, and it has been revised.

The Conclusion section has been checked.

309-313

Funding  "the authors did research with the help of volunteers". Who were the volunteers?

The missing information has been provided.

350-351

Reviewer 2 Report

The manuscript presents an important topic. The authors have described the topic very well, however, the methods and results section need improvement and clarifications.

I have the following specific comments to the authors:

  1. In the methods section, under study design, it is not clear how the participants were recruited actually. I think you should explain how you selected your participants, and how you decided your sample size? Also, the response rate and how many non-respondents ..
  2. Please clarify the methods of collecting the data, was it an online questionnaire? or a paper form sent to them? how you collected the filled questionnaires?
  3. You mentioned that the study did not need ethical review. Please explain why and on what basis?
  4.  It would be good to elaborate a bit on the sections of the questionnaire in the methods section.
  5. The level of knowledge needs more clarification. What scale you used what were the questions asked, and how you calculated the level?

Author Response

Dear Reviewer,

Thank you for presenting us with an opportunity to submit the revised version of the manuscript entitled ‘Knowledge on the complementary feeding of infants older than six months among mothers following vegetarian and traditional diets.

We greatly appreciate the time and effort taken by the Reviewer to review our manuscript. We have addressed all issues indicated in the review report, and we believe that the revised version will meet the Journal’s requirements. The manuscript was checked by a professional translator.

Please find attached our responses to the Reviewer’s comments. The manuscript has been corrected, and all changes are highlighted in orange font.

Yours sincerely,

Malgorzata Kostecka

Thank you very much for your insightful review. We greatly appreciate the time and effort taken to review our manuscript and we agree that the proposed changes will contribute to the improvement of our manuscript. We hope you will find our improvements appropriate and comprehensive.

In the methods section, under study design, it is not clear how the participants were recruited actually. I think you should explain how you selected your participants, and how you decided your sample size? Also, the response rate and how many non-respondents ..

Thank you for this valuable observation, the description has been completed.

73-74

77-80

Please clarify the methods of collecting the data, was it an online questionnaire? or a paper form sent to them? how you collected the filled questionnaires?

Thank you for this valuable observation, the description has been completed

90-92

You mentioned that the study did not need ethical review. Please explain why and on what basis?

The consent of the bioethics committee is not required in the case of anonymous questionnaire studies on child nutrition, without the analysis of laboratory tests or the possibility of identifying the respondents. All participants were informed about the study and the long-term use of data.

The relevant information is included in the Ethics approval and consent to participate subsection.

357-359

 It would be good to elaborate a bit on the sections of the questionnaire in the methods section.

Thank you for this valuable observation, the description has been completed

95-96,98

The level of knowledge needs more clarification. What scale you used what were the questions asked, and how you calculated the level?

Thank you for this valuable observation, the description has been completed

102-117
